# Separation Results between Fixed-Kernel and Feature-Learning Probability Metrics

**Carles Domingo-Enrich**
Courant Institute of Mathematical Sciences (NYU)
cd2754@nyu.edu

**Youssef Mroueh**
IBM Research AI
mroueh@us.ibm.com

## Abstract

Several works in implicit and explicit generative modeling empirically observed that feature-learning discriminators outperform fixed-kernel discriminators in terms of the sample quality of the models. We provide separation results between probability metrics with fixed-kernel and feature-learning discriminators using the function classes $\mathcal{F}_2$ and $\mathcal{F}_1$ respectively, which were developed to study overparametrized two-layer neural networks. In particular, we construct pairs of distributions over hyper-spheres that can not be discriminated by fixed kernel ($\mathcal{F}_2$) integral probability metric (IPM) and Stein discrepancy (SD) in high dimensions, but that can be discriminated by their feature learning ($\mathcal{F}_1$) counterparts. To further study the separation we provide links between the $\mathcal{F}_1$ and $\mathcal{F}_2$ IPMs with sliced Wasserstein distances. Our work suggests that fixed-kernel discriminators perform worse than their feature learning counterparts because their corresponding metrics are weaker.

## 1 Introduction

The field of generative modeling, whose aim is to generate artificial samples of some target distribution given some true samples of it, is broadly divided into two types of models (Mohamed and Lakshminarayanan, 2017): explicit generative models, which involve learning an estimate of the log-density of the target distribution which is then sampled (e.g. energy-based models), and implicit generative models, where samples are generated directly by transforming some latent variable (e.g. generative adversarial networks (Goodfellow et al., 2014), normalizing flows (Rezende and Mohamed, 2015)).

Several works have observed experimentally that both in implicit and in explicit generative models, using 'adaptive' or 'feature-learning' function classes as discriminators yields better generative performance than 'lazy' or 'kernel' function classes. Within implicit models, Li et al. (2017) show that generative moment matching networks (GMMN) generate significantly better samples for the CIFAR-10 and MNIST datasets when using maximum mean discrepancy (MMD) with learned instead of fixed features. For a related method and in a similar spirit, Santos et al. (2019) show that for image generation, fixed-feature discriminators are only successful when we take an amount of features exponential in the intrinsic dimension of the dataset. Genevay et al. (2018) study implicit generative models with the Sinkhorn divergence as discriminator, and they also show that other than for simple datasets like MNIST, learning the Wasserstein cost is crucial for good performance.

As to explicit models, Grathwohl et al. (2020) train energy-based models with a Stein discrepancy based on neural networks and show improved performance with respect to kernel classes. Chang et al. (2020) show that Stein variational gradient descent (SVGD) fails in high dimensions, and that learning the kernel helps. Given the abundant experimental evidence, the aim of this work is to provide some theoretical results that showcase the advantages of feature-learning over kernel discriminators. For

35th Conference on Neural Information Processing Systems (NeurIPS 2021).

the sake of simplicity, we compare the discriminative behavior of two function classes $\mathcal{F}_1$ and $\mathcal{F}_2$, arising from infinite-width two-layer neural networks with different norms penalties on its weights (Bach, 2017). $\mathcal{F}_1$ displays an adaptive behavior, while $\mathcal{F}_2$ is an RKHS which consequently has a lazy behavior. Namely, our main contributions are:

(i) We construct a sequence of pairs of distributions over hyperspheres $\mathbb{S}^{d-1}$ of increasing dimensions, such that the $\mathcal{F}_2$ integral probability metric (IPM) between the pair decreases exponentially in the dimension, while the $\mathcal{F}_1$ IPM remains high.

(ii) We construct a sequence of pairs of distributions over $\mathbb{S}^{d-1}$ such that the $\mathcal{F}_2$ Stein discrepancy (SD) between the pair decreases exponentially in the dimension, while the $\mathcal{F}_1$ SD remains high.

(iii) We prove polynomial upper and lower bounds between the $\mathcal{F}_1$ IPM and the max-sliced Wasserstein distance for distributions over Euclidean balls. For a class $\tilde{\mathcal{F}}_2$ related to $\mathcal{F}_2$, we prove similar upper and lower bounds between the $\tilde{\mathcal{F}}_2$ IPM and the sliced Wasserstein distance for distributions over Euclidean balls.

Our findings reinforce the idea that generative models with kernel discriminators have worse performance because their corresponding metrics are weaker and thus unable to distinguish between different distributions, especially in high dimensions.

## 2 Related work

A recent line of research has studied the question of how neural networks compare to kernel methods, with a focus on supervised learning problems. Bach (2017) shows the approximation benefits of the $\mathcal{F}_1$ space for adapting to low-dimensional structures compared to the (kernel) space $\mathcal{F}_2$; an analysis that we leverage. The function space $\mathcal{F}_1$ was also studied by Ongie et al. (2019); Savarese et al. (2019); Williams et al. (2019), which focus on the ReLU activation function. More recently, several works showed that wide neural networks trained with gradient methods may behave like kernel methods in certain regimes (see, e.g., Jacot et al., 2018). Examples of works that compare 'active/feature-learning' and 'kernel/lazy' regimes for supervised learning include Chizat and Bach (2020); Ghorbani et al. (2019); Wei et al. (2020); Woodworth et al. (2020), and Domingo-Enrich et al. (2021) for energy-based models. We are not aware of any works that study how feature-learning function classes and kernel classes differ as discriminators for IPMs or Stein discrepancies.

It turns out that the $\mathcal{F}_2$ integral probability metric that we study is in fact MMD for certain kernels that often admit a closed form (Roux and Bengio, 2007; Cho and Saul, 2009; Bach, 2017). MMDs are probability metrics that were first introduced by Gretton et al. (2007, 2012) for kernel two-sample tests, and that have enjoyed ample success with the advent of deep-learning-based generative modeling as discriminating metrics: Li et al. (2015) and Dziugaite et al. (2015) introduced GMMN, which differ from GANs in that the discriminator network is replaced by a fixed-kernel MMD. Li et al. (2017) introduces an improvement on GMMN by using the MMD loss on learned features. From this viewpoint, our separation results in Sec. 5 can be interpreted as instances in which a given fixed-kernel MMD provably has less discriminative power than adaptive discriminators.

Other related work includes the Stein discrepancy literature. Stein's method (Stein, 1972) dates to the 1970s. Gorham and Mackey (2015) introduced a computational approach to compute the Stein discrepancy in order to assess sample quality. Later, Chwialkowski et al. (2016), Liu et al. (2016) and Gorham and Mackey (2017) introduced the more practical kernelized Stein discrepancy (KSD) for goodness-of-fit tests. Liu and Wang (2016) introduced SVGD, the first method to use the KSD to obtain samples from a distribution. Barp et al. (2019) employed KSD to train parametric generative models, and Grathwohl et al. (2020) trained models replacing KSD by a neural-network-based SD.

Our work also touches on sliced and spiked Wasserstein distances. Sliced Wasserstein distances were introduced first by Kolouri et al. (2016); Kolouri et al. (2019). Spiked Wasserstein distances, which are a generalization, were studied later by Paty and Cuturi (2019), and they also appear in Niles-Weed and Rigollet (2019) as a good statistical estimator. Nadjahi et al. (2020) and Lin et al. (2021) have studied statistical properties of sliced and spiked Wasserstein distances, respectively.

## 3 Framework

### 3.1 Notation

If $V$ is a normed vector space, we use $\mathcal{B}_V(\beta)$ to denote the closed ball of $V$ of radius $\beta$, and $\mathcal{B}_V := \mathcal{B}_V(1)$ for the unit ball. If $K$ denotes a subset of the Euclidean space, $\mathcal{P}(K)$ is the set of Borel probability measures, $\mathcal{M}(K)$ is the space of finite signed Radon measures and $\mathcal{M}^+(K)$ is the set of finite positive Radon measures. If $\gamma$ is a signed Radon measure over $K$, then $\|\gamma\|_{\mathrm{TV}}$ is the total variation (TV) norm of $\gamma$. Throughout the paper, and unless otherwise specified, $\sigma : \mathbb{R} \to \mathbb{R}$ denotes a generic non-linear activation function. We use $(\cdot)_+ : \mathbb{R} \to \mathbb{R}$ to denote the ReLu activation, defined as $(x)_+ = \max\{x, 0\}$. $\tau$ denotes the uniform probability measure over a space that depends on the context. We use $\mathbb{S}^d$ for the $d$-dimensional hypersphere and $\log$ for the natural logarithm.

### 3.2 Overparametrized two-layer neural network spaces

**Feature-learning regime.** We define $\mathcal{F}_1$ as the Banach space of functions $f : K \to \mathbb{R}$ such that for some $\gamma \in \mathcal{M}(\mathbb{S}^d)$, for all $x \in K$ we have $f(x) = \int_{\mathbb{S}^d} \sigma(\langle \theta, x \rangle) \, d\gamma(\theta)$ for some signed Radon measure $\gamma$ (Bach, 2017). The norm of $\mathcal{F}_1$ is defined as $\|f\|_{\mathcal{F}_1} = \inf \left\{ \|\gamma\|_{\mathrm{TV}} \mid f(\cdot) = \int_{\mathbb{S}^d} \sigma(\langle \theta, \cdot \rangle) \, d\gamma(\theta) \right\}$.

**Kernel regime.** We define $\mathcal{F}_2$ as the (reproducing kernel) Hilbert space of functions $f : K \to \mathbb{R}$ such that for some absolutely continuous $\rho \in \mathcal{M}(\mathbb{S}^d)$ with $\frac{d\rho}{d\tau} \in \mathcal{L}^2(\mathbb{S}^d)$ (where $\tau$ is the uniform probability measure over $\mathbb{S}^d$), we have that for all $x \in K$, $f(x) = \int_{\mathbb{S}^d} \sigma(\langle \theta, x \rangle) \, d\rho(\theta)$. The norm of $\mathcal{F}_2$ is defined as $\|f\|_{\mathcal{F}_2}^2 = \inf \left\{ \int_{\mathbb{S}^d} h(\theta)^2 \, d\tau(\theta) \mid f(\cdot) = \int_{\mathbb{S}^d} \sigma(\langle \theta, \cdot \rangle) \, h(\theta) \, d\tau(\theta) \right\}$. As an RKHS, the kernel of $\mathcal{F}_2$ is

$$k(x, y) = \int_{\mathbb{S}^d} \sigma(\langle x, \theta \rangle) \sigma(\langle y, \theta \rangle) d\tau(\theta). \tag{1}$$

Such kernels admit closed form expressions for different choices of activation functions, among which ReLu (Roux and Bengio, 2007; Cho and Saul, 2009; Bach, 2017).

Remark that since $\int |h(\theta)| d\tau(\theta) \leq (\int h(\theta)^2 \, d\tau(\theta))^{1/2}$ by the Cauchy-Schwarz inequality, we have $\mathcal{F}_2 \subset \mathcal{F}_1$. In particular, when $\sigma$ is the ReLu unit, Bach (2017) shows that two-layer networks with a single neuron belong to $\mathcal{F}_1$ but not to $\mathcal{F}_2$, and their $L^2$ approximations in $\mathcal{F}_2$ have exponentially high norm in the dimension. Informally, one should understand $\mathcal{F}_1$ as the space of two-layer networks where both the input layer and output layer parameters are trained, in the limit of an infinite number of neurons. On the other hand, $\mathcal{F}_2$ is the space of infinite-width two-layer networks where only the output layer parameters are trained while the input layer parameters are sampled uniformly on the sphere and kept fixed.

## 4 $\mathcal{F}_1$ and $\mathcal{F}_2$ Integral Probability Metrics

Let $K$ be a subset of $\mathbb{R}^{d+1}$. Integral probability metrics (IPM) are pseudometrics on $\mathcal{P}(K)$ of the form

$$d_{\mathcal{F}}(\mu, \nu) = \sup_{f \in \mathcal{F}} \mathbb{E}_{x \sim \mu} f(x) - \mathbb{E}_{x \sim \nu} f(x),$$

where $\mathcal{F}$ is a class of functions from $K$ to $\mathbb{R}$.

**$\mathcal{F}_2$ IPM or $\mathcal{F}_2$ MMD.** One possible choice for $\mathcal{F}$ is the unit ball $\mathcal{B}_{\mathcal{F}_2}$ of $\mathcal{F}_2$. Since $\mathcal{F}_2$ is an RKHS with kernel $k$, the corresponding IPM is in fact a maximum mean discrepancy (MMD) (Gretton et al., 2007) and it can be shown (Lemma 1 in App. A) to take the form

$$d_{\mathcal{B}_{\mathcal{F}_2}}^2(\mu, \nu) = \int_{\mathbb{S}^d} \left( \int_K \sigma(\langle x, \theta \rangle) d(\mu - \nu)(x) \right)^2 d\tau(\theta).$$

Notice that for any feature $\theta \in \mathbb{S}^d$, $\mathbb{E}_{x \sim p} \sigma(\langle x, \theta \rangle)$ can be seen as a generalized moment of $p$. $d_{\mathcal{B}_{\mathcal{F}_2}}$ can be seen as the $L^2$ distance between generalized moments of $\mu$ and $\nu$ as functions of $\theta \in \mathbb{S}^d$.

$\mathcal{F}_1$ **IPM.** An alternative choice for $\mathcal{F}$ is the unit ball $\mathcal{B}_{\mathcal{F}_1}$ of $\mathcal{F}_1$. The IPM for the unit ball of $\mathcal{F}_1$ can be developed (Lemma 2 in App. A) into

$$d_{\mathcal{B}_{\mathcal{F}_1}}(\mu, \nu) = \sup_{\theta \in \mathbb{S}^d} \left| \int_K \sigma(\langle x, \theta \rangle) d(\mu - \nu)(x) \right|. \tag{2}$$

Observe that $d_{\mathcal{B}_{\mathcal{F}_1}}$ is the $L^\infty$ distance between generalized moments of $\mu$ and $\mu$ as functions of $\theta \in \mathbb{S}^d$. That is, instead of averaging over features, all the weight is allocated to the feature at which the generalized moment difference is larger.

We will provide separate results for two interesting choices for $K$: (i) for $K = \mathbb{S}^d$, we obtain neural network discriminators without bias term which are amenable to analysis using the theory of spherical harmonics; and (ii) for $K = \mathbb{R}^d \times \{1\}$, we obtain neural networks discriminators with a bias term which is encoded by the last component (notice that probability measures over $\mathbb{R}^d$ can be mapped trivially to probability measures over $\mathbb{R}^d \times \{1\}$). We will write $\mathcal{F}_1(K)$ or $\mathcal{F}_2(K)$ for specific $K$ when it is not clear by the context.

A function $f : \mathbb{R} \to \mathbb{R}$ is $\alpha$-positive homogeneous function if for all $r \geq 0, x \in \mathbb{R}, f(rx) = r^\alpha f(x)$. One-dimensional $\alpha$-positive homogeneous functions can be written in a general form as

$$f(x) = a(x)_+^\alpha + b(-x)_+^\alpha. \tag{3}$$

where $a, b \in \mathbb{R}$ are arbitrary. When the activation function $\sigma$ is $\alpha$-positive homogeneous, Theorem 1 shows that the $\mathcal{F}_1$ and $\mathcal{F}_2$ IPMs are distances when $K = \mathbb{R}^d \times \{1\}$ if $a, b$ fulfill a certain condition which is satisfied by the ReLu activation, but they are *not* distances when $K = \mathbb{S}^d$. See Theorem 6 and Theorem 7 in App. B for the proof.

**Theorem 1.** *For any non-negative integer $\alpha$, let $\sigma : \mathbb{R} \to \mathbb{R}$ be an $\alpha$-positive homogeneous activation function of the form* (3). *If $(-1)^\alpha a - b \neq 0$ and $K = \mathbb{R}^d \times \{1\}$, both the $\mathcal{F}_1$ and $\mathcal{F}_2$ IPMs are distances on $\mathcal{P}(K)$. If $K = \mathbb{S}^d$, both the $\mathcal{F}_1$ and $\mathcal{F}_2$ IPMs are not distances on $\mathcal{P}(K)$, as there exist pairs of different measures for which the IPMs evaluate to zero.*

In other words, Theorem 1 states that certain fixed-kernel and feature-learning infinite neural networks with RELU or leaky RELU non-linearity, yield distances when we include a bias term, but not when the inputs lie in a hypersphere. This result sheds light on when the "neural net distance" introduced by Arora et al. (2017) is indeed a distance.

## 5 Separation between the $\mathcal{F}_1$ and $\mathcal{F}_2$ IPMs

In this section for each dimension $d \geq 2$, we construct a pair of probability measures $\mu_d, \nu_d$ over $\mathcal{P}(\mathbb{S}^{d-1})$ such that the $\mathcal{F}_1$ IPM between $\mu_d$ and $\nu_d$ stays constant along the dimension, while the $\mathcal{F}_2$ IPM decreases exponentially.

**Legendre harmonics and Legendre polynomials.** Let $e_d \in \mathbb{R}^d$ be the $d$-th vector of the canonical basis. There is a unique homogeneous harmonic polynomial $L_{k,d}$ of degree $k$ over $\mathbb{R}^d$ such that: (i) $L_{k,d}(Ax) = L_{k,d}(x)$ for all orthogonal matrices that leave $e_d$ invariant, and (ii) $L_{k,d}(e_d) = 1$. This polynomial receives the name of *Legendre harmonic*, and its restriction to $\mathbb{S}^{d-1}$ is indeed a spherical harmonic of order $k$. If we express an arbitrary $\xi_{(d)} \in \mathbb{S}^{d-1}$ as $\xi_{(d)} = te_d + (1 - t^2)^{1/2} \xi_{(d-1)}$, where $\xi_{(d-1)} \perp e_d$, we can define the *Legendre polynomial* of degree $k$ in dimension $d$ as $P_{k,d}(t) := L_{k,d}(\xi_{(d)})$ by the invariance of $L_{k,d}$ (it is not straightforward that $P_{k,d}(t)$ is a polynomial on $t$, see Sec. 2.1.2 of Atkinson and Han (2012)). Conversely, $L_{k,d}(x) = P_{k,d}(\langle e_d, x \rangle)$ for any $x \in \mathbb{S}^{d-1}$, and by homogeneity, $L_{k,d}(x) = \|x\|^k P_{k,d}(\langle e_d, x \rangle / \|x\|)$ for any $x \in \mathbb{R}^d$. Legendre polynomials can also be characterized as the orthogonal sequence of polynomials on $[-1, 1]$ such that $P_{k,d}(1) = 1$ and $\int_{-1}^1 P_{k,d}(t) P_{l,d}(t)(1 - t^2)^{\frac{d-3}{2}} dt = 0$, for $k \neq l$.

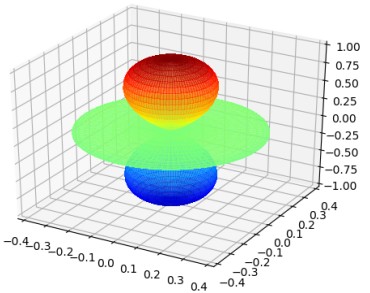 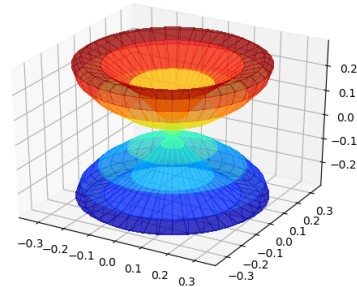

Figure 1: 3D polar plot representing the densities of the measures $\mu_d$ (left) and $\nu_d$ (right), for the choices $d = 3$, $k = 4$. In each direction, the distance from the origin to the surface is proportional to the density of the measure.

**The pair $\mu_d$ and $\nu_d$.** We define $\mu_d$ and $\nu_d$ as the probability measures over $\mathbb{S}^{d-1}$ with densities

$$
\begin{aligned}
\frac{d\mu_d}{d\lambda} &= \begin{cases} \frac{\gamma_{k,d} L_{k,d}(x)}{|\mathbb{S}^{d-1}|} & \text{if } L_{k,d}(x) > 0 \\ 0 & \text{if } L_{k,d}(x) \leq 0 \end{cases}, \\
\frac{d\nu_d}{d\lambda} &= \begin{cases} 0 & \text{if } L_{k,d}(x) > 0 \\ \frac{-\gamma_{k,d} L_{k,d}(x)}{|\mathbb{S}^{d-1}|} & \text{if } L_{k,d}(x) \leq 0 \end{cases}.
\end{aligned} \tag{4}
$$

for some $k \geq 2$ and some $\gamma_{k,d} \geq 0$, where $\lambda$ is the Hausdorff measure over $\mathbb{S}^{d-1}$. Namely,

**Proposition 1.** *If we choose $\gamma_{k,d} = 2 \left( \int_{\mathbb{S}^{d-1}} |L_{k,d}(x)| \, d\tau(x) \right)^{-1}$, then $\mu_d$ and $\nu_d$ are probability measures.*

Figure 1 shows a representation of the measures $\mu_d, \nu_d$ for $d = 3, k = 4$, where one can see that they allocate mass in different regions of the sphere. We are now ready to state our separation result, which is proved in App. D.

**Theorem 2.** *Let $\sigma : \mathbb{R} \to \mathbb{R}$ be an activation function that is bounded in $[-1, 1]$. For any $d \geq 2$ and $k \geq 1$, if we set $\gamma_{k,d}$ as in Proposition 1 we have that*

$$
d_{\mathcal{B}_{\mathcal{F}_1}}(\mu_d, \nu_d) = \frac{2 \left| \int_{-1}^{1} P_{k,d}(t) \sigma(t)(1 - t^2)^{\frac{d-3}{2}} \, dt \right|}{\int_{-1}^{1} |P_{k,d}(t)|(1 - t^2)^{\frac{d-3}{2}} dt}, \tag{5}
$$

*and*

$$
\frac{d_{\mathcal{B}_{\mathcal{F}_1}}(\mu_d, \nu_d)}{d_{\mathcal{B}_{\mathcal{F}_2}}(\mu_d, \nu_d)} = \sqrt{N_{k,d}} = \sqrt{\frac{(2k + d - 2)(k + d - 3)!}{k!(d - 2)!}}, \tag{6}
$$

*where $N_{k,d}$ is the dimension of the space of spherical harmonics of order $k$ over $\mathbb{S}^{d-1}$. That is,*

$$
\log\left( \frac{d_{\mathcal{B}_{\mathcal{F}_1}}(\mu_d, \nu_d)}{d_{\mathcal{B}_{\mathcal{F}_2}}(\mu_d, \nu_d)} \right) = \frac{1}{2} \left( k \log\left( \frac{k + d - 3}{k} \right) + (d - 2) \log\left( \frac{k + d - 3}{d - 2} \right) \right) + O(\log(k + d)). \tag{7}
$$

From (7) we see that choosing the parameter $k$ of the same order as $d$, $d_{\mathcal{B}_{\mathcal{F}_1}}(\mu_d, \nu_d)$ is exponentially larger than $d_{\mathcal{B}_{\mathcal{F}_2}}(\mu_d, \nu_d)$ in the dimension $d$. Equation (5) holds regardless of the choice of the activation function $\sigma$, and decreases very slowly in $d$ for the ReLu activation, as shown in Figure 2. This result suggests that in high dimensions there exist high frequency densities that can be distinguished by feature-learning IPM discriminators but not by their fixed-kernel counterpart, and that may explain the differences in generative modeling performance for GMMN and Sinkhorn divergence (Sec. 1).

The key idea for the proof of Theorem 2 is that the Legendre harmonics $L_{k,d}$ have constant $L^\infty$ norm equal to 1 (see equation (26) in App. C), but their $L^2$ norm decreases as $1/N_{k,d}$ (see equation (32) in App. D). The proof boils down to relating $d_{\mathcal{B}_{\mathcal{F}_1}}(\mu_d, \nu_d)$ to the $L^\infty$ norm of $L_{k,d}$, and $d_{\mathcal{B}_{\mathcal{F}_2}}(\mu_d, \nu_d)$ to its $L^2$ norm.

# 6 Separation between $\mathcal{F}_1$ and $\mathcal{F}_2$ Stein discrepancies

The arguments to derive the separation result in Sec. 5 can be leveraged to obtain a similar separation for the Stein discrepancy, which helps explain why for Stein discrepancy energy-based models (EBMs) and SVGD feature learning yields improved performance.

## 6.1 Stein operator and Stein discrepancy

As shown by Domingo-Enrich et al. (2021), for a probability measure $\nu$ on the sphere $\mathbb{S}^{d-1}$ with a continuous and almost everywhere differentiable density $\frac{d\nu}{d\tau}$, the *Stein operator* $\mathcal{A}_\nu : \mathbb{S}^{d-1} \to \mathbb{R}^{d \times d}$ is defined as

$$(\mathcal{A}_\nu h)(x) = \left( \nabla \log \left( \frac{d\nu}{d\tau}(x) \right) - (d-1)x \right) h(x)^\top + \nabla h(x), \qquad (8)$$

for any $h : \mathbb{S}^{d-1} \to \mathbb{R}^d$ that is continuous and almost everywhere differentiable, where $\nabla$ denotes the Riemannian gradient. That is, for any $h : \mathbb{S}^{d-1} \to \mathbb{R}^d$ that is continuous and almost everywhere differentiable, the Stein identity holds: $\mathbb{E}_\nu[(\mathcal{A}_\nu h)(x)] = 0$.

If $\mathcal{H}$ is a class of functions from $\mathbb{S}^{d-1}$ to $\mathbb{R}^d$, the Stein discrepancy (Gorham and Mackey, 2015; Liu et al., 2016) for $\mathcal{H}$ is a non-symmetric functional defined on pairs of probability measures over $K$ as

$$\mathrm{SD}_{\mathcal{H}}(\nu_1, \nu_2) = \sup_{h \in \mathcal{H}} \mathbb{E}_{\nu_1}[\mathrm{Tr}(\mathcal{A}_{\nu_2} h(x))]. \qquad (9)$$

When $\mathcal{H} = \mathcal{B}_{\mathcal{H}_0^d} = \{(h_i)_{i=1}^d \in \mathcal{H}_0^d \mid \sum_{i=1}^d \|h_i\|_{\mathcal{H}_0}^2 \le 1\}$ for some reproducing kernel Hilbert space (RKHS) $\mathcal{H}_0$ with kernel $k$ with continuous second order partial derivatives, there exists a closed form for the problem (9) and the corresponding object is known as kernelized Stein discrepancy (KSD) Liu et al. (2016); Gorham and Mackey (2017). When the domain is $\mathbb{S}^{d-1}$, the KSD takes the following form (Lemma 5, Domingo-Enrich et al. (2021)):

$$\mathrm{KSD}(\nu_1, \nu_2) = \mathrm{SD}^2_{\mathcal{B}_{\mathcal{H}_0^d}}(\nu_1, \nu_2) = \mathbb{E}_{x,x' \sim \nu_1}[u_{\nu_2}(x, x')],$$

where we have $u_\nu(x, x') = (s_\nu(x) - (d-1)x)^\top (s_\nu(x') - (d-1)x')k(x, x') + (s_\nu(x) - (d-1)x)^\top \nabla_{x'} k(x, x') + (s_\nu(x') - (d-1)x')^\top \nabla_x k(x, x') + \mathrm{Tr}(\nabla_{x,x'} k(x, x'))$, and we use $\tilde{u}_\nu(x, x')$ to denote the sum of the first three terms (remark that the fourth term does not depend on $\nu$). Here we have used the notation $s_\nu(x) = \nabla \log(\frac{d\nu}{d\tau}(x))$, which is known as the score function.

## 6.2 Separation result

We show a separation result between the two cases:

- $\mathcal{F}_1$ Stein discrepancy: $\mathcal{H} = \mathcal{B}_{\mathcal{F}_1^d} = \{(h_i)_{i=1}^d \in \mathcal{F}_1^d \mid \sum_{i=1}^d \|h_i\|_{\mathcal{F}_1}^2 \le 1\}$. This discriminator set initially appeared as a particular configuration in the framework of Huggins and Mackey (2018), and its statistical properties for energy based model training were later studied by Domingo-Enrich et al. (2021).

- $\mathcal{F}_2$ Stein discrepancy: $\mathcal{H} = \mathcal{B}_{\mathcal{F}_2^d} = \{(h_i)_{i=1}^d \in \mathcal{F}_2^d \mid \sum_{i=1}^d \|h_i\|_{\mathcal{F}_2}^2 \le 1\}$. Since $\mathcal{F}_2$ is an RKHS, this corresponds to a KSD with the kernel $k$. However, particular care must be taken in checking that the kernel $k$ has continuous second order partial derivatives, which might not always be the case (i.e. with $\alpha = 1$).

**The pair $\mu_d$ and $\nu_d$.** For $d \geq 2$, we set $\mu_d$ to be the uniform Borel probability measure over $\mathbb{S}^{d-1}$. We define $\nu_d$ as the probability measure over $\mathbb{S}^{d-1}$ with density

$$\frac{d\nu_d}{d\lambda}(x) = \frac{\exp\left(\gamma_{k,d} L_{k,d}(x)\right)}{\int_{\mathbb{S}^{d-1}} \exp\left(\gamma_{k,d} L_{k,d}(x)\right) d\lambda(x)} \tag{10}$$

for some $\gamma_{k,d} \in \mathbb{R}$ that we will specify later on and some $k \geq 2$.

**Theorem 3.** *Let $\sigma : \mathbb{R} \to \mathbb{R}$ be an $\alpha$-positive homogeneous activation function of the form* (3) *such that $a + (-1)^{k+1}b \neq 0$. For all $k \geq 1$, $d \geq 2$ we can choose $\gamma_{k,d} \in [-1,1]$ such that $SD_{\mathcal{B}_{\mathcal{F}_1^d}}(\mu_d, \nu_d) = 1$ and*

$$\frac{SD_{\mathcal{B}_{\mathcal{F}_1^d}}(\mu_d, \nu_d)}{SD_{\mathcal{B}_{\mathcal{F}_2^d}}(\mu_d, \nu_d)} \geq \frac{\frac{k(d+k-3)}{\alpha+1}}{\sqrt{\frac{2}{N_{k,d}}\left(k(k+d-2)\left(\frac{d+\alpha-2}{\alpha+1}\right)^2 + \left(\frac{k(d+k-3)}{\alpha+1}\right)^2\right)}} \tag{11}$$

*That is,*

$$\log\left(\frac{SD_{\mathcal{B}_{\mathcal{F}_1^d}}(\mu_d, \nu_d)}{SD_{\mathcal{B}_{\mathcal{F}_2^d}}(\mu_d, \nu_d)}\right) \geq \frac{1}{2}\left(k\log\left(\frac{k+d-3}{k}\right) + (d-2)\log\left(\frac{k+d-3}{d-2}\right)\right) + O(\log(k+d)) \tag{12}$$

As in Theorem 2, from (12) we see that choosing the parameter $k$ of the same order as $d$, $SD_{\mathcal{B}_{\mathcal{F}_1^d}}(\mu_d, \nu_d)$ is exponentially larger than $SD_{\mathcal{B}_{\mathcal{F}_2^d}}(\mu_d, \nu_d)$ in the dimension $d$. This result suggests that in high dimensions there exist high frequency densities that can be distinguished by feature-learning Stein Discrepancy discriminators but not by their fixed-kernel counterpart, and that may explain the differences in generative modeling performance for Stein discrepancy EBMs and SVGD (Sec. 1).

# 7 Bounds of $\mathcal{F}_1$ and $\mathcal{F}_2$ IPMs by sliced Wasserstein distances

$\mathcal{F}_1$ and $\mathcal{F}_2$ IPMs measure differences of densities by slicing the input space and then maximizing (resp. averaging) the appropriate quantities. Max-sliced and sliced Wasserstein distances work, which have been studied by several works, work in an analogous fashion; one projects the distributions onto one-dimensional subspaces, and then maximizes or averages over the subspaces. Unlike the Wasserstein distance, which has been used for generative models such as WGAN (Arjovsky et al., 2017) but whose estimation suffers from the curse of dimensionality, max-sliced and sliced Wasserstein enjoy parametric estimation rates which make them more suitable as discriminators.

The goal of this section is to show that $\mathcal{F}_1$ IPMs are equivalent to max-sliced Wasserstein distances up to a constant power, while sliced Wasserstein distances are similarly equivalent to a fixed-kernel IPM with a kernel that is slightly different from the $\mathcal{F}_2$ kernel. These bounds are helpful to get a quantitative understanding of how strong feature-learning and fixed-kernel IPMs are, and provide a novel bridge between sliced optimal transport and generative modeling discriminators.

## 7.1 Spiked and sliced Wasserstein distances

Throughout this section $k$ denotes an integer such that $1 \leq k \leq d$. The Stiefel manifold $\mathcal{V}_k$ is the set of matrices $U \in \mathbb{R}^{k \times d}$ such that $UU^\top = I_{k \times k}$ (i.e. the rows of $U$ are orthonormal). We define the *$k$-dimensional projection robust $p$-Wasserstein distance* between $\mu, \nu \in \mathcal{P}(\mathbb{R}^d)$ as

$$\overline{\mathcal{W}}_{p,k}(\mu, \nu)^p = \max_{U \in \mathcal{V}_k} \min_{\pi \in \Gamma(\mu,\nu)} \int \|Ux - Uy\|^p d\pi(x,y), \tag{13}$$

where $\Gamma(\mu, \nu)$ denotes the set of couplings between $\mu, \nu$, i.e. of measures $\mathcal{P}(K \times K)$ with projections $\mu$ and $\nu$. This is the distance studied by Niles-Weed and Rigollet (2019) as a good estimator for the Wasserstein distance for a certain class of target densities with low dimensional structure.

The *integral k-dimensional projection robust p-Wasserstein distance* between $\mu, \nu \in \mathcal{P}(\mathbb{R}^d)$ is defined as

$$\underline{\mathcal{W}}_{p,k}(\mu,\nu)^p = \int_{\mathcal{V}_k} \left( \min_{\pi \in \Gamma(\mu,\nu)} \int \|Ux - Uy\|^p d\pi(x,y) \right) d\tau(U), \tag{14}$$

where $\tau$ is the uniform measure over $\mathcal{V}_k$. Nadjahi et al. (2020) studied statistical aspects of this distance in the case in which $k = 1$, while Lin et al. (2021) considers the case with general $k$. Notice trivially that $\overline{\mathcal{W}}_{p,k}(\mu,\nu) \geq \underline{\mathcal{W}}_{p,k}(\mu,\nu)$.

Sliced Wasserstein distances are spiked Wasserstein distances with $k = 1$, but they were studied first chronologically (Bonneel et al., 2014; Kolouri et al., 2016; Kolouri et al., 2019). Namely, the *sliced Wasserstein distance* is the integral 1-dimensional projection robust Wasserstein distance $\underline{\mathcal{W}}_{p,k}$, and the *max-sliced Wasserstein distance* is the 1-dimensional projection robust Wasserstein distance $\overline{\mathcal{W}}_{p,k}$. Some arguments are easier for the case $k = 1$ because the Stiefel manifold is the sphere $\mathbb{S}^{d-1}$.

## 7.2 Results

We prove in Theorem 4 that for $K = \{x \in \mathbb{R}^d | \|x\|_2 \leq 1\} \times \{1\}$, for which the $\mathcal{F}_1$ space corresponds to overparametrized two-layer neural networks with bias, the $\mathcal{F}_1$ IPM can be upper and lower-bounded by the projection robust Wasserstein distance $\overline{\mathcal{W}}_{1,k}(\mu,\nu)$ up to a constant power (not depending on the dimension).

**Theorem 4.** *Let $\delta > 0$ be larger than a certain constant depending on $k$ and $\alpha$. Let $\sigma(x) = (x)_+^\alpha$ be the $\alpha$-th power of the ReLu activation function, where $\alpha$ is a non-negative integer. Let $\mu, \nu$ be Borel probability measures with support included in $\{x \in \mathbb{R}^d | \|x\|_2 \leq 1\} \times \{1\}$. Let $d_{\mathcal{B}_{\mathcal{F}_1}}$ be as defined in (2) and $\overline{\mathcal{W}}_{1,k}$ as defined in (13). Then,*

$$\delta \overline{\mathcal{W}}_{1,k}(\mu,\nu) \geq \delta d_{\mathcal{B}_{\mathcal{F}_1}}(\mu,\nu) \geq \overline{\mathcal{W}}_{1,k}(\mu,\nu) - 2C(k,\alpha)\delta^{-\frac{1}{\alpha+(k-1)/2}} \log(\delta), \tag{15}$$

*where $C(k,\alpha)$ is a constant that depends only on $k$ and $\alpha$. If we optimize the lower bound in (15) with respect to $\delta$, we obtain $\overline{\mathcal{W}}_{1,k}(\mu,\nu) \geq d_{\mathcal{B}_{\mathcal{F}_1}}(\mu,\nu) \geq \tilde{\Omega}(\overline{\mathcal{W}}_{1,k}(\mu,\nu)^{\alpha+\frac{k+1}{2}})$ where $\tilde{\Omega}$ hides log factors.*

While for the $\mathcal{F}_2$ IPM the link with the sliced Wasserstein distance is not straightforward, it can be established when we switch from uniform $\tau$ to an alternative feature measure $\tilde{\tau}$. We define the class $\tilde{\mathcal{F}}_2$ of functions $\mathbb{R}^d \to \mathbb{R}$ as the RKHS associated with the following kernel

$$\tilde{k}(x,y) = \int_{\mathbb{S}^d} \sigma(\langle (x,1), \theta \rangle) \sigma(\langle (y,1), \theta \rangle) \, d\tilde{\tau}(\theta) =$$

$$\frac{1}{\pi} \int_{\mathbb{S}^{d-1}} \int_{-1}^{1} \sigma\left(\langle (x,1), (\sqrt{1-t^2}\xi, t) \rangle\right) \sigma\left(\langle (y,1), (\sqrt{1-t^2}\xi, t) \rangle\right) (1-t^2)^{-1/2} \, dt \, d\tau_{(d-1)}(\xi).$$

**Proposition 2.** ($\tilde{\tau}$ **as a rescaling of uniform measure**) *The measure $d\tilde{\tau}(\sqrt{1-t^2}\xi, t) = \frac{1}{\pi}(1 - t^2)^{-1/2} dt \, d\tau_{(d-1)}(\xi)$ is a probability measure. For comparison, the uniform measure over $\mathbb{S}^d$ can be written as $d\tau(\sqrt{1-t^2}\xi, t) = \frac{\Gamma((d+1)/2)}{\sqrt{\pi}\Gamma(d/2)}(1 - t^2)^{\frac{d-1}{2}} dt \, d\tau_{(d-1)}(\xi)$.*

That is, $\mathcal{F}_2$ and $\tilde{\mathcal{F}}_2$ are both fixed-kernel spaces with a similar kernel. They differ only in the weighing measure of the kernel; all the expressions which are valid in the $\mathcal{F}_2$ setting are also valid for $\tilde{\mathcal{F}}_2$ if we replace $\tau$ by $\tilde{\tau}$. In analogy with the $\mathcal{F}_2$ IPM, the $\tilde{\mathcal{F}}_2$ IPM is given below.

$$d^2_{\mathcal{B}_{\tilde{\mathcal{F}}_2}}(\mu,\nu) = \int_{\mathbb{S}^d} \left( \int_K \sigma(\langle x, \theta \rangle) d(\mu - \nu)(x) \right)^2 d\tilde{\tau}(\theta). \tag{16}$$

Analogously to Theorem 4, Theorem 5 establishes that the $\tilde{\mathcal{F}}_2$ IPM is upper and lower-bounded by the sliced Wasserstein distance $\underline{\mathcal{W}}_{1,1}(\mu,\nu)$ up to a constant power (not depending on the dimension). The reason to introduce the space $\tilde{\mathcal{F}}_2$ is that in the proof, the argument that makes the connection with the sliced Wasserstein distance requires the base measure of the kernel to be $\tilde{\tau}$ and does not work for $\tau$. However, we do not imply that a similar result for the $\mathcal{F}_2$ IPM is false.

**Theorem 5.** *Let $\delta > 0$ be larger than a certain constant depending on $k$ and $\alpha$. Let $\sigma(x) = (x)_+^\alpha$ be the $\alpha$-th power of the ReLu activation function, where $\alpha$ is a non-negative integer. Let $\mu, \nu$ be Borel probability measures with support included in $\{x \in \mathbb{R}^d \,|\, \|x\|_2 \le 1\} \times \{1\}$. Let $d_{\mathcal{B}_{\tilde{\mathcal{F}}_2}}$ be as defined in (16) and $\underline{\mathcal{W}}_{1,1}$ as defined in (14). Then,*

$$\delta d_{\tilde{\mathcal{F}}_2}^{2/3}(\mu, \nu) \ge \left(\frac{5}{12\pi\alpha 2^{\alpha/2}}\right)^{1/3} \left(\underline{\mathcal{W}}_{1,1}(\mu, \nu) - 2C(1, \alpha)\delta^{-\frac{1}{\alpha}} \log(\delta)\right). \tag{17}$$

*and $\pi d_{\mathcal{B}_{\tilde{\mathcal{F}}_2}}^2(\mu, \nu) \le \underline{\mathcal{W}}_{1,1}(\mu, \nu)$. If we optimize the lower bound in (17) with respect to $\delta$, we obtain $d_{\tilde{\mathcal{F}}_2}^{2/3}(\mu, \nu) \ge \tilde{\Omega}(\underline{\mathcal{W}}_{1,1}(\mu, \nu)^{1+\alpha})$.*

## 8    Experiments

To validate and clarify our findings, we perform experiments of the settings studied Sec. 5, Sec. 6 and Sec. 7. We use the ReLu activation function $\sigma(x) = (x)_+$, although remark that the results of Sec. 5 hold for a generic activation function, and the results of Sec. 6 and Sec. 7 hold for non-negative integer powers of the ReLu activation. The empirical estimates in the plots are detailed in App. G. They are averaged over 10 repetitions; the error bars show the maximum and minimum.

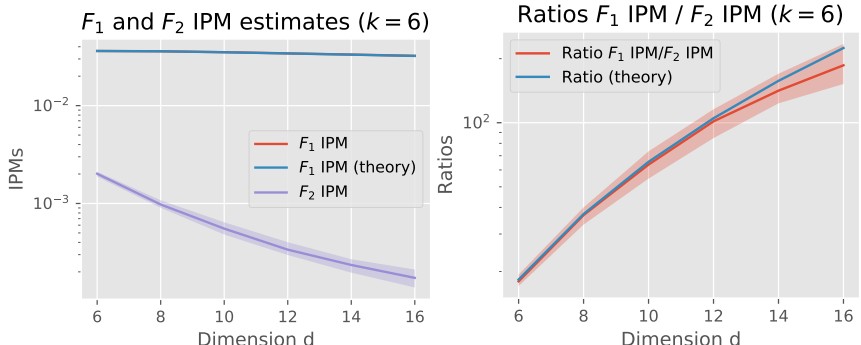

Figure 2: $\mathcal{F}_1$ and $\mathcal{F}_2$ IPM estimates for the pairs $\mu_d$ and $\nu_d$ defined in (4) for $k = 6$ and varying dimension $d$. (Left) The blue and red curves (superposed) show two different estimates of the $\mathcal{F}_1$ IPM. The purple curve shows estimates for the $\mathcal{F}_2$ IPM. (Right) The blue curve shows the theoretical ratio between the $\mathcal{F}_1$ and the $\mathcal{F}_2$ IPMs (see equation (6)). The red curve shows an empirical estimate of the ratio obtained by dividing the IPM estimates. 4400 million samples of $\mu_d$ and $\nu_d$ are used.

**Separation between $\mathcal{F}_1$ and $\mathcal{F}_2$ IPMs.** Figure 2 shows $\mathcal{F}_1$ and $\mathcal{F}_2$ IPM estimates for the pairs $\mu_d$ and $\nu_d$ defined in (4) for the Legendre polynomial of degree $k = 6$ and varying dimension $d$, and its ratios. We observe that while the $\mathcal{F}_1$ IPM remains nearly constant in the dimension, the $\mathcal{F}_2$ IPM experiences a significant decrease. The ratios between IPMs closely track those predicted by our Theorem 2, the mismatch being due to the overestimation of the $\mathcal{F}_2$ IPM caused by statistical errors. We were constrained in the values of $k$ and $d$ that we could choose; when the $\mathcal{F}_2$ IPM is small, which is the case when $k$ and/or $d$ are large, we need a high number of samples from the distributions $\mu_d, \nu_d$ to make the statistical error smaller than $d_{\mathcal{B}_{\mathcal{F}_2}}(\mu_d, \nu_d)$ and get a good estimate.

**Separation between $\mathcal{F}_1$ and $\mathcal{F}_2$ SDs.** Figure 3 shows $\mathcal{F}_1$ and $\mathcal{F}_2$ SD estimates for the pairs $\mu_d$ and $\nu_d$ defined in equation Subsec. 6.2 for the Legendre polynomial of degree $k = 5$ and varying dimension $d$, and its ratios. The fact that the empirical ratio is significantly above the theoretical lower bound indicates that our lower bound (although exponential) is not tight. This can be guessed by looking at the slackness in the inequalities of Lemma 10 and Lemma 11.

$\mathcal{F}_1$, $\mathcal{F}_2$, $\tilde{\mathcal{F}}_2$ **IPMs versus max-sliced and sliced Wasserstein.** Figure 4 shows several metrics between a standard multivariate Gaussian and a Gaussian with unit variance in all directions except for one of smaller variance $0.1$, in varying dimensions. We observe that while the $\mathcal{F}_1$ IPM and the max-sliced Wasserstein distance are constant, the $\mathcal{F}_2$, $\tilde{\mathcal{F}}_2$ IPMs and the sliced Wasserstein distance

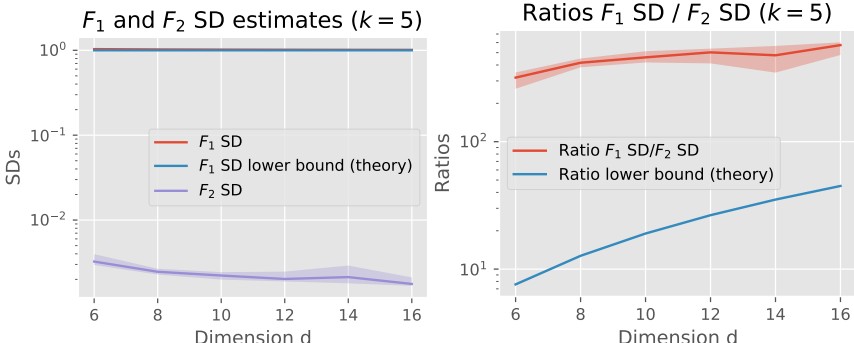

Figure 3: $\mathcal{F}_1$ and $\mathcal{F}_2$ SD estimates for the pairs $\mu_d$ and $\nu_d$ defined in Subsec. 6.2 for $k = 5$ and varying dimension $d$. (Left) The red curve shows an empirical estimate of the $\mathcal{F}_1$ SD, the blue curve shows a theoretical lower bound (Lemma 10 in App. E) on the $\mathcal{F}_1$ SD, the purple curve shows an estimate of the $\mathcal{F}_2$ SD. (Right) The blue curve represents the theoretical lower bound on the ratio between the $\mathcal{F}_1$ and the $\mathcal{F}_2$ SDs (see equation (11)), while the red curve shows an empirical estimate of the ratio obtained by dividing the SD estimates. 30 million samples are used.

decrease. For high dimensions they match the corresponding distances between two datasets of standard multivariate Gaussian, which means that the statistical noise precludes discrimination in these metrics.

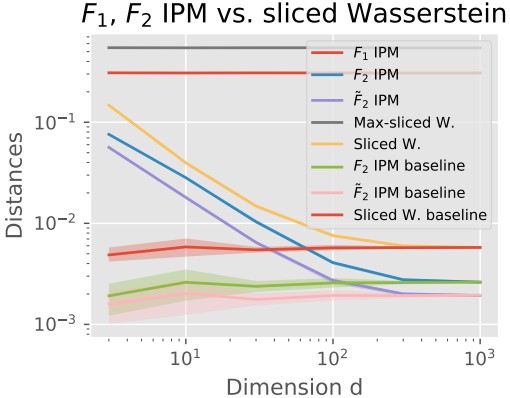

Figure 4: For varying dimension $d$, we plot $\mathcal{F}_1$, $\mathcal{F}_2$, $\tilde{\mathcal{F}}_2$ IPM, sliced and max-sliced Wasserstein estimates between a standard multivariate Gaussian and a Gaussian with unit variance in all directions except for one of smaller variance $0.1$. The estimates are computed using 100000 samples of each distribution. For comparison, the same estimates are shown between a standard multivariate Gaussian and itself, using two different sets of 100000 samples.

## 9   Conclusions and discussion

We have shown pairs of distributions over hyperspheres for which the $\mathcal{F}_1$ IPM and SD are exponentially larger than the $\mathcal{F}_2$ IPM and SD. In parallel, we have also provided links between the $\mathcal{F}_1$ IPM and max-sliced Wasserstein distance, and between the $\tilde{\mathcal{F}}_2$ IPM and the sliced Wasserstein distance. The densities of the distributions constructed in Sections Sec. 5 and Sec. 6 are based on Legendre harmonics of increasing degree. Keeping in mind that spherical harmonics are the Fourier basis for $L^2(\mathbb{S}^{d-1})$ (in the sense that they constitute an orthonormal basis of eigenvalues of the Laplace-Beltrami operator), one can infer a simple overarching idea from our constructions: '$\mathcal{F}_1$ discriminators are better than $\mathcal{F}_2$ discriminators at telling apart distributions whose densities have only high frequency differences. It would be interesting to develop this intuition into a more general theory. Another avenue of future work is to understand how deep discriminators perform versus shallow ones, in analogy with the work of Eldan and Shamir (2016) for regression.

**Acknowledgements.**   We thank Joan Bruna for useful discussions. CD acknowledges partial support by "la Caixa" Foundation (ID 100010434), under agreement LCF/BQ/AA18/11680094.

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
