# OpenReview forum: "Separation Results between Fixed-Kernel and Feature-Learning Probability Metrics"
_NeurIPS.cc/2021/Conference — NeurIPS 2021 Oral_

### Official Review · Reviewer_7tHG · 2021-07-10

**Rating:** 8
**Confidence:** 3

**Summary:**

This paper argues that discriminators based on learned features have significantly more discrimination power than discriminators based on fixed features. It does so by defining two function classes that can be interpreted as infinite width one hidden layer neural networks, one corresponding to the learned, or adaptive regime (F_1), and another corresponding to the fixed, or kernel regime (F_2). Various interesting separation and characterization results are shown between F_1 and F_2, using practically relevant metrics.

**Limitations And Societal Impact:**

Yes.

**Main Review:**

The paper is written very cleanly, and it has a clear message and demonstration of that message: learned features have greater discrimination power than a fixed kernel. The theoretical nature of the paper complements previous empirical demonstrations of this message.

The results also appear very clean. I am not familiar with constructions involving spherical harmonics, but within this area, they seem natural. A sequence of pairs of probability distributions defined by the Legendre harmonic is defined which exhibits an exponential gap between the F_1 IPM and the F_2 IPM, and the authors obtain closed form expressions for the IPMs, which is nice. I also found it interesting that Theorem 1 sheds light on when a previously defined "neural net distance" is actually a distance. The separation in terms of Stein discrepancy is also interesting since the Stein discrepancy is another popular metric used in discrimination tasks, and the within polynomial factor correspondences between F_1 IPM and max-sliced Wasserstein distance and between F_2 IPM and sliced Wasserstein distance is an interesting, unifying characterization.

Since the paper is so clean and has useful results, I vote to accept. I seems like it would be a useful reference to other researchers in this area.

A question that I have about the technical novelty of the paper is, how different is the usage of spherical harmonics here from the Bach 2017 reference. What did Bach 2017 use spherical harmonics for, and how similar are any constructions involving them between this paper and that one?
A note: I did not see s_{nu} defined anywhere, in the definition of the kernelized stein discrepancy.

**Time Spent Reviewing:**

5

---

> ### Author Response · Authors · 2021-08-06
> **Response to Reviewer 7tHG**
>
> We thank the reviewer for their remarks, and are pleased about their opinion on the paper.
>
> * **“[...] how different is the usage of spherical harmonics here from the Bach 2017 reference. What did Bach 2017 use spherical harmonics for, and how similar are any constructions involving them between this paper and that one?”**
>
> The use of spherical harmonics in Bach [2017] is as an analytical tool to characterize the spaces of interest ($F_1,F_2$), but not as a way to construct counterexamples as in our paper.
>
> The proofs of our Theorems 2 and 3 are  based on spherical harmonics in a constructive way, and  the arguments are very different from the ones in Bach [2017] (except that we also use the Funk-Hecke formula in equation 31). The key idea for the proof of Theorem 2 is that the Legendre harmonics $L_{k,d}$ have constant $L^{\infty}$ norm equal to 1 (see equation 26), but their $L^2$ norm decreases as $ 1/\sqrt{N_{k,d}} $ (see equation 32). The proof of Theorem 3 has a similar flavor but the details are more involved.
> Bach [2017] uses spherical harmonics to compute the eigenvalues of the space $F_2$ as an RKHS. With that, he goes on to show that functions with smoothness of the order of the dimension belong to $F_2$ (his Proposition 2), and also that Lipschitz functions may be approximated by $F_2$ balls at a certain rate that depends on the dimension (his Proposition 3, which we reproduce in our Lemma 14). He uses this last result to bound the approximation error of functions with a certain low-dimensional structure, and dealing with the statistical error via a Rademacher complexity bound, in his Subsections 5.2 and 5.3 he obtains generalization bounds for regression problems in which such functions are the targets.
>
> In our paper, we also make use of the approximation result of Lipschitz functions by functions in $F_2$ in the proofs of Theorem 4 and Theorem 5. In Theorem 4, as explained in lines 739-741, the max-sliced k-Wasserstein distance can be written in terms of the matrix $U^*$ and Lipschitz function $f^*$ on $\mathbb{R}^k$ where the supremum is achieved. Thus, the proof follows from approximating $f^*$.
>
> * **“A note: I did not see s_{nu} defined anywhere, in the definition of the kernelized stein discrepancy.”**
>
> As mentioned in an answer to Reviewer tbhZ, $s_{\nu}$ denotes the score function for the probability measure $\nu$, and it is defined as $s_{\nu}(x) = \nabla \log (\frac{d\nu}{d\tau}(x)) $. We forgot to define it, but will add the definition in lines 195-196.

---

### Official Review · Reviewer_tbhZ · 2021-07-10

**Rating:** 9
**Confidence:** 3

**Summary:**

The authors study two classes of functions, F1 and F2, which are related to
"infinite-width" two-layer neural nets and fixed-kernel RKHSes. They show
that in the regime where d goes to infinity, the sensitivity of MMDs based
on F2 decreases exponentially comapred to F1 for specific pairs of measures
on the hypersphere S^d. The authors corroborrate this theoretical find with
some simulations.

**Limitations And Societal Impact:**

The authors did not include an impact on the societal impact of their work
and should add one!

**Main Review:**

While mostly a theoretical contribution, the work here seems like a concrete
addition to understanding the differences between F1 and F2. There are also
some nice connections made to the sliced and max-sliced Wasserstein metrics
which establishes some nice bounds for these quantities.

Originality:
The constructive examples given in the paper seem quite carefully
constructed. While this work builds upon much other recent work, the core
theorems are original and elucidating.

Quality:
The claims in the paper look correct. Outside of a couple ambiguous parts,
the paper was well written.

Clarity:
Despite the difficulty of the subject, the paper was quite pleasant to read.

Significance:
While the paper may not have any immediate practical import, it does made a
theoretical contribution to an important problem in discriminator theory. In
order to bolster this area, I would suggest that the authors expound a bit
more on F1 and F2 and how they related to modern NNs/GANs in a bit more
detail at the beginning of the paper.

Detailed Comments:

L110-111: It might be worthwhile to expound a bit on what "single units"
are, as this is a key fact for claims that F1 is better than F2.

L155: "as x = r zeta_{(d)} with zeta_{(d)}," should belong to S^(d-1)
Theorem 2: The wording here is a little ambiguous. Does "we can choose k >=
0 and gamma_{k,d} in [-1, 1]" mean that we can choose any such values and
(5-7) will hold? Or does this mean we need a speciifc choice of values for
these to hold? Moreover, why does (5) not depend on gamma_{k,d}?

L194: Where is s_{nu} defined?

L276: Can you explain exactly what "by statistical errors" means?

**Time Spent Reviewing:**

3

---

> ### Author Response · Authors · 2021-08-06
> **Response to Reviewer tbhZ**
>
> We thank the reviewer for the comments, and are pleased about their opinion on the paper.
>
> * **“I would suggest that the authors expound a bit more on F1 and F2 and how they related to modern NNs/GANs in a bit more detail at the beginning of the paper.”**
>
> This is indeed something that can be made clearer. On the one hand, one should understand the space $F_1$ as the space of  two-layer neural networks where both the input layer and output layer parameters are trained, in the limit of an infinite number of neurons. On the other hand, the space $F_2$ is the space of two-layer neural networks where only the output layer parameters are trained while the input layer parameters are sampled uniformly on the sphere and kept fixed, in the limit of an infinite number of neurons. From this perspective, the only difference between $F_1$ and $F_2$ is whether we train the input layer weights (i.e. the features) or not. In terms of similarities with more complex models, we can see $F_1$ as an abstraction of fully connected networks or CNNs where feature learning takes places, whereas $F_2$ is closer to other neural network kernel regimes that have been studied extensively in recent years, such as the neural tangent kernel or NTK (which is also an RKHS albeit with different kernel). In the context of GANs, the $F_1$ space corresponds to discriminators with learned features, the space  $F_2$ corresponds to discriminators based on predefined feature matching such as pretrained VGG19 features in image generation.     We will add an explanation like this one in Sec. 3.2, because it will tie the $F_1$ and $F_2$ spaces better with the feature learning dichotomy discussed earlier in the paper.
>
> * **“L110-111: It might be worthwhile to expound a bit on what "single units" are, as this is a key fact for claims that F1 is better than F2.”**
>
> By “single units” we refer to two-layer neural networks with a single neuron (although the sentence would also be true if we said “a finite number of neurons”). The wording will be improved to make it clearer.
>
> * **“L155: "as x = r zeta_{(d)} with zeta_{(d)}," should belong to S^(d-1)"**
>
> Yes, it should be “with $\zeta_{(d)} \in \mathbb{S}^{d-1}$”. The typo will be corrected.
>
> * **“Theorem 2: [...] Does "we can choose k >= 0 and gamma_{k,d} in [-1, 1]" mean that we can choose any such values and (5-7) will hold? Or does this mean we need a specific choice of values for these to hold?”**
>
> Our wording needs to be made precise. It should say: “For any $d \geq 2$ and $k \geq 1$, if we set $\gamma_{k,d}$ as in Proposition 1, then [...]”. The condition $k \geq 2$ mentioned in line 162 is a typo and will be removed, and we will also add the condition $k \geq 1$ in Proposition 1 for clarity (because that is where it is needed, in particular in equation 29 in App. D). We need $d \geq 2$ to make use of equation 26 in the appendix.
>
> * **“L194: Where is s_{nu} defined?”**
>
> $s_{\nu}$ denotes the score function for the probability measure $\nu$, and it is defined as $s_{\nu}(x) = \nabla \log (\frac{d\nu}{d\tau}(x)) $. We forgot to define it, but will add the definition in lines 195-196.
>
> * **“L276: Can you explain exactly what "by statistical errors" means?”**
>
> This question is somewhat related to the second answer that we give to Reviewer tnLw. When we compute IPMs between measures using their samples, we have to deal with a statistical error going as 1/sqrt(no. samples). To get a good estimation of the $F_2$ IPM (which is critical to obtain a good estimation of the ratio of the IPMs), we need the statistical error of $d_{B_{F_2}}(\mu, \nu)$ to be smaller than its magnitude, and for that we need a number of samples large enough. Otherwise, we overestimate $d_{B_{F_2}}(\mu, \nu)$ and consequently we underestimate the ratio.
>
> * **“The authors did not include an impact on the societal impact of their work and should add one!”**
>
> We will add a brief discussion on the societal impact of our work in Section 9. As mentioned in our answer to question 1(c), since our work is of theoretical nature, there are no direct societal implications. However, we agree that it is a good idea to include this comment in the main text.

---

> > ### Comment · Reviewer_tbhZ · 2021-08-22
> > **Rebuttal response**
> >
> > I thank the authors and other reviewers for their responses. I still believe the contribution of this work is beneficial, especially given that it provides a constructive example for the different convergence rates between F1 and F2. Reviewer e39f mentioned
> >
> > "the story that feature learning is beneficial as opposed to fixed kernel is not very surprising not only empirically but also theoretically if the reader is familiar with the F_1 vs F_2 literature."
> >
> > While this seems reasonable based on the NN interpretations of F1 and F2 and how those interpretations fare empirically, I myself was not so sure that there would be a nice theoretical justification that these were inherently different. If the reviewer provides some reference that helped develop their intuition here, that would be both good as a reference for the paper and useful for me to update my score.
> >
> > That being said, while there isn't a great deal of empirical work in the paper, I still find this to be a significant result worth of publication and will be sticking with my score for now.

---

> > > ### Comment · Reviewer_e39f · 2021-08-23
> > > **references on F1**
> > >
> > > Showing the inherent difference theoretically is basically the goal of the literature on F1 spaces - e.g. papers/talks by Robert Nowak (linking it to locally adaptive splines e.g. https://www.youtube.com/watch?v=HNFhSm1-rtM) and Bach '17 (different generalization bounds, less explicit though) are a starting point. Now all of them try to make a similar point (that was what I meant by my comment) but their results do offer different perspectives on what kind of differences might appear so that is not my main criticism of the paper. I would find it important to see why this particular theoretical "difference" that is worked out is relevant to explaining phenomena the broader ML community might care about by adding some experiments that link it back to practice.

---

> > > > ### Author Response · Authors · 2021-08-24
> > > > **Thanks and response to comments**
> > > >
> > > > We appreciate the comment of Reviewer tbhZ  and share their view on the contributions of the paper. As mentioned by Reviewer e39f, we acknowledge that the previous works by Bach and Nowak flesh out the theoretical differences between adaptive and lazy function spaces (we already cite Bach [2017] but will cite  Nowak’s papers as well). However, to our knowledge none of these works study the differences between these spaces in the context of discriminators for generative modeling.  We summarize here our main contributions:  1)  we formally define $F_1$/$F_2$ integral probability metrics, 2) we prove when they define proper metrics 3)  we provide separation results between $F_1$/$F_2$ IPMs and provide similar results for  Stein discrepancies, 4)  we further deepen the link between $F_1$ and $F_2$ IPM  and sliced optimal transport. As far as we know, none of these have been explored in previous works in the $F_1$/$F_2$ literature.

---

### Official Review · Reviewer_tnLw · 2021-07-17

**Rating:** 5
**Confidence:** 2

**Summary:**

The authors propose theoretical results about separation between probability metrics with fixed-kernel and feature-learning discriminators in overparameterized two-layer neural network spaces.

**Limitations And Societal Impact:**

Yes

**Main Review:**

The authors propose theoretical results about separation between probability metrics with fixed-kernel and feature-learning discriminators in overparameterized two-layer neural network spaces.

In Theorem 2 and 3, the authors derive the ratio between feature-learning $F_1$ and fixed-kernel $F_2$ integral probability metrics (IPM), and for $F_1$ and $F_2$ Stein respectively. The authors illustrate that on the hyper-spheres, the $F_1$ IPM/Stein has much larger values than $F_2$ IPM/Stein which seems to infer that the separation of $F_1$ IPM/Stein is better than $F_2$ IPM/Stein (line 170-175 and line 214-217).
+ Do the $F_1$ IPM/Stein have different range values than $F_2$ IPM/Stein? (since it is unclear whether their values have a same range)
+ Does the separation relate to the range value of a discrepancy d? since with k>1, the discrepancy kd is k-times larger than discrepancy d, does it imply the separation of discrepancy kd is better than that of discrepancy d?
--> It is better in case the authors give a more formal definition of "separation". How does the "separation" relate to the value of a discrepancy between probability measures?

Some other concerns:
+ in line 108, $F_2 \subset F_1$, could the authors elaborate this property with more details? Can one use this property for $F_1$ and $F_2$ IPM directly since the IPM is maximum over $F$ space?
+ For results in Section 7.2, what is the relation between $F_2$ and $\tilde{F}_2$?
+ In experiments, how does the degree of Legendre polynomial affect the results?

Some minor concerns:
+ For the sliced Wasserstein, I think it is better to cite the work of Rabin et al, 2011 which is the first paper about sliced-Wasserstein.

References:
+ Julien Rabin, Gabriel Peyré, Julie Delon, and Marc Bernot. Wasserstein barycenter and its application to texture mixing. In International Conference on Scale Space and Variational Methods in Computer Vision, pages 435–446, 2011.

------After the rebuttal-----
I thank the authors for the rebuttal. I think that it is better in case the authors improve the clarification about those raised points (I feel that there are several implications that make it hard for the readers to access its contributions.
From the discussions with other reviewers, the theoretical part is good, but the contribution can be further improved with some well-designed experiments to illustrate the proposed theoretical results.

**Time Spent Reviewing:**

8 hours

---

> ### Author Response · Authors · 2021-08-06
> **Response to Reviewer tnLw**
>
> We thank the reviewer for their thoughtful remarks.
>
> * **“Do the $F1$ IPM/Stein have different range values than $F2$ IPM/Stein?”**
>
>  Being pseudometrics, the ranges of $F_1$ and $F_2$ IPMs for measures on $\mathbb{S}^{d-1}$ are intervals whose lower end is zero. The $F_1$ and $F_2$ Stein discrepancies for measures on $\mathbb{S}^{d-1}$ also have ranges of the same form. When we talk about separation, we do not mean that the ranges are disjoint, but rather that there exist a distinct pairs of measures such that the $F_1$ IPM is larger than the $F_2$ IPM by a factor that is exponential in the dimension $d$. In a sense that, in a hypothesis testing scenario,  $F_1$ IPM is able to distinguish if these two measures are distinct , while $F_2$ IPM does not.
>
> * **“since with k>1, the discrepancy kd is k-times larger than discrepancy d, does it imply the separation of discrepancy kd is better than that of discrepancy d?“**
>
> This question raises an interesting point that we touch on in lines 284-286, but that we will put more emphasis on. In a hypothetical world where we had exact access to the measures $ \mu,\nu $, we could effectively multiply the $F_2$ IPM by a constant factor exponential in the dimension $d$ and no harm would be done. However, we typically want to compute IPMs between measures using their samples, which means we have to deal with a statistical error going as 1/sqrt(no. samples). When the F_2 IPM takes very small values, the statistical error is much larger than the signal unless the number of samples is exponential in $d$. Thus, we will not obtain a good estimate, and this is so even if we multiply the discrepancy by a constant (the statistical error will be multiplied as well).
> This is in fact the intuition behind statistical hypothesis testing. Suppose that the null hypothesis is that $\mu = \nu$. Suppose that the tests we consider are whether the $F_1/F_2$ IPMs are above or below a certain threshold. The type I error or level of the test is the probability that the null hypothesis is rejected despite $\mu = \nu$, and the type II error is the probability that the null hypothesis is accepted despite $\mu \neq \nu$. For a fixed $\mu = \nu$, if $d_{B_{F_2}}(\mu, \nu)$ is smaller than the statistical error it will not be possible to keep both the type I and type II errors low for any threshold. See A Kernel Two-Sample Test by Gretton, Borgwardt, Rasch, Schölkopf and Smola [2012] for more details on statistical hypothesis testing with MMD.
>
> * **“in line 108, $ F_2 \subset F_1 $, could the authors elaborate this property with more details? Can one use this property for $F_1$ and $F_2$ IPM directly since the IPM is maximum over F space?”**
>
> The inclusion $ F_2 \subset F_1 $ holds because whenever a function on $K$ can be written as $ f(x) = \int_{\mathbb{S}^{d}} \sigma(\langle x, \theta \rangle) h(\theta) d\tau(\theta) $ where $h$ with finite $L^2$ norm  , we can write $ f(x) = \int_{\mathbb{S}^{d}} \sigma(\langle x, \theta \rangle) d\mu(\theta) $, where the Radon measure $\mu$ has density $d\mu/d\tau = h$. Moreover, the total variation norm of $\mu$ given by $\int_{\mathbb{S}^{d}} |h(\theta)| d\tau(\theta) \leq (\int_{\mathbb{S}^{d}} |h(\theta)| d\tau(\theta))^{1/2}$, where the last equality holds by the Cauchy-Schwarz inequality as indicated. The inclusion appears in page 7 of Bach [2017], which is why do not provide exhaustive details, but we will do a better job at guiding the reader to the reference. Regarding the IPMs, the fact that  the $F_1$ norm is less or equal than the $F_2$ norm for any function in $F_2$ implies that unit ball of $F_2$ is included in the unit ball of $F_1$. Consequently the $F_1$ IPM is larger or equal than the $F_2$ IPM (because it is the supremum over a larger class of functions).
>
> * **“For results in Section 7.2, what is the relation between $F_2$ and $tildeF_2$?”**
>
>  We will try to make the introduction of the space $tildeF_2 $ clearer. While $F_2$ can be seen as the RKHS with kernel $k(x,y) = \int_{\mathbb{S}^{d}} \sigma(\langle \theta, x \rangle) \sigma(\langle \theta, x \rangle) d\tau(\theta)$ where $\tau$ is the uniform measure over the hypersphere (lines 101-102), $tildeF_2$ is the RKHS we obtain when we replace $\tau$ by the probability measure $tilde \tau$, which has density described in lines 255-256.
> That is, $F_2$ and $tildeF_2$ are both fixed-kernel spaces, and they differ in the weighing measure of the kernel. The fact that they behave similarly is exemplified in Figure 3. $tildeF_2$ is introduced because in Theorem 5 the argument that makes the connection with the sliced Wasserstein distance requires the base measure of the kernel to be $tilde \tau$. It does not work with the uniform measure $\tau$, which does not mean that the result is false.
> Since the experiments are a validation of the theory, we know with certainty the dependency in the degree $k$ of the Legendre polynomial with the dimension: it is given by Theorem 2 for the IPMs and Theorem 3 for the Stein discrepancy (a tight result in the former case; a bound in the latter). As mentioned in lines 170-171 and 214-215, the exponential separation in $d$ takes place when $k$ is of the order of $d$.
>
> * **“For the sliced Wasserstein, I think it is better to cite the work of Rabin et al, 2011 [...]”**
>
> We missed this reference and will give it its due credit.

---

### Official Review · Reviewer_e39f · 2021-07-20

**Rating:** 5
**Confidence:** 3

**Summary:**

This paper provides a theoretical justification for why feature learning (theoretical proxy for that is the F_1 space) may be crucial as opposed to learning with fixed features (theoretical proxy for that is the RKHS with NTK or F_2 space) when using statistical pseudometrics that are computed using the sup over function spaces (e.g. IPM pseudometrics and Stein discrepancy) to train a generative model to match a given training distribution. In particular, they construct sequences of pairs of distributions for which they can prove that as d increases, F_1 pseudometrics yield much higher “distance” between the distributions than F_2 pseudometrics. In particular, the mathematical exercise suggests that distributions that differ in high frequency components benefit more from feature learning.

**Limitations And Societal Impact:**

purely theoretical work

**Main Review:**

Strengths:

The results and the presentation of the mathematical results are clean and solid (however I have not had a chance to check the proofs in detail). The story that feature learning is beneficial as opposed to fixed kernel is not very surprising not only empirically but also theoretically if the reader is familiar with the F_1 vs F_2 literature. Still, showing it for generative modeling explicitly is novel and could be valuable.

Constructive comments:

In order to make the results more impactful within the ML community however, the results seem to directly suggest an experiment to link back the theoretical insights to the practical motivation: i.e. can you design an experiment that illustrates that the insight from the distribution pairs (different high frequency components) that are indistinguishable as d-> infty for F_2, carries over to the image scenarios that motivate this study?For example, something that immediately comes to mind is that you could remove/project out the high frequency components and see if the FID without feature learning then gets better

Moreover, the presentation could be much improved. Here is some more detailed feedback

- The figures are quite poorly presented - use different linestyle. The two red curves in Figure 3 - which is which?

- For IPM your theoretical prediction is quite accurate. For Stein discrepancy - it’s not clear that the ratio is actually increasing with d empirically albeit being high in general. I’m missing a discussion on these Stein discrepancy simulations in general

- Section 7 and appendix generally seem relatively underdeveloped - the relevance of assumptions and implications of Theorem 5 should be directly discussed and interpreted in order for the reader to know what to focus on while parsing it and perhaps the proof. Furthermore, the switch from F_2 to Ftilde_2 is not well motivated - what do we learn about F_2 by looking at Ftilde_2 instead?

- You mention setting k=d near the discussion of the implications of the theorem and later in the experiments set k constant for all d - discussion would help

- Dimensionality - In sections 3, 4, and 7, the inputs are considered to lie in a subset K of the (d+1)-dimensional space (l. 114), and the paper considers statistical distances between distributions \mu and \nu over K, for two special cases: K=R^d \times {1} or K=S^d. However in sections 5 and 6, the distributions \mu_d and \nu_d are defined over S^{d-1}. Might be useful to keep it consistent

Minor comments

- l. 178: define the acronym EBM, and add the relevant references again
- line 171 it seems like eq (6) should read eq (5)
- eq (33) is the same as (6) - was a bit confusing in the main text


**Time Spent Reviewing:**

6

---

> ### Author Response · Authors · 2021-08-06
> **Response to Reviewer e39f**
>
> We thank the reviewer for their thoughtful remarks, which will surely help in making the paper clearer.
>
> * **“can you design an experiment that illustrates that the insight from the distribution pairs (different high frequency components) that are indistinguishable as d-> infty for F_2, carries over to the image scenarios that motivate this study?”**
>
> This is something that would be highly interesting, and related to the comment that we make in lines 294-296. In the present work we have decided to keep the focus on the theoretical results, but we plan to pursue this line of work in the future and acknowledge that obtaining such results would be of value to practitioners in their choice of architectures and training methods.
>
> * **“For example, something that immediately comes to mind is that you could remove/project out the high frequency components and see if the FID without feature learning then gets better”**
>
> Thanks for the suggestion! For $F_2$ an infinite number of features will be still  important in order to train a generative model  with small FID score as  it can be seen from figure 7 in https://arxiv.org/pdf/1904.02762.pdf , matching feature on all layers is  important to generate natural images. Figure 7 of the aforementioned paper, suggests that  low frequency content can be effectively generated by matching on a smaller number of features using $F_2$. We will consider your suggestion to analyse $F_2$ in high dimension.
>
> * **“The figures are quite poorly presented - use different linestyle. The two red curves in Figure 3 - which is which?”**
>
> The $F_1$ IPM is the second horizontal line starting from the top; the sliced Wasserstein baseline is the other line. Instead of so many colors, thanks for the suggestion,  we will use different line styles (dashed lines, dotted lines...) to make this plot clearer.
>
> * **“For IPM your theoretical prediction is quite accurate. For Stein discrepancy - it’s not clear that the ratio is actually increasing with d empirically albeit being high in general. I’m missing a discussion on these Stein discrepancy simulations in general”**
>
> The fact that the empirical ratio $F_1$ SD/$F_2$ SD is significantly above the theoretical lower bound indicates that our lower bound (although exponential) is not tight. This can be guessed by looking at the slackness in the inequalities of Lemma 12 (see lines 689-691) and Lemma 13 (see Line 707). It would be interesting to get values of the empirical ratio for other choices of $k$ and $d$, but unfortunately, as we mention in our fifth answer to this review (“You mention…”), we are constrained in the values of $k$ and $d$ for which we can get good estimates of the $F_2$ SD with a reasonable amount of samples from $\mu_d, \nu_d$.
> We will add this discussion in the paragraph “Separation between $F_1$ and $F_2$ SDs”.
>
> * **“Section 7 and appendix generally seem relatively underdeveloped - the relevance of assumptions and implications of Theorem 5 should
> be directly discussed and interpreted in order for the reader to know what to focus on while parsing it and perhaps the proof.”**
>
> __Assumption:__ The main assumptions are that the activation function chosen is a power of the ReLu, and that the support of the distributions $\mu, \nu$ is included in {$x \in \mathbb{R}^{d} | \|x\|_2 \leq 1$} $\times$ {$1$}. The product with $\{1\}$ is just to capture neural networks with a bias term, but one can think of such measures as supported in the Euclidean unit ball.
>
> __Implications:__ The main idea of Section 7 is to show that the $F_1$ and $F_2$ IPMs are equivalent to the max-sliced and sliced 1-Wasserstein distances up to a constant power respectively. Sliced Wasserstein distances and spiked Wasserstein distances have been popular in applied optimal transport as exemplified by the works that we cite. Their main advantage over regular Wasserstein distances is that their estimation from empirical measures does not suffer from the curse of dimensionality. Like the $F_1$ IPM, in max-sliced 1-Wasserstein distances the direction for which the two distributions are most different is selected. And as in the $F_2$ IPM, in sliced 1-Wasserstein distances what matters is the average value over all directions. In practice the main implication of this theorem is that using $F_1$ and $F_2$  IPM can be seen as a proxy to sliced (max and average resp.) optimal transport.
>
> __Proof:__  As we mention in our first answer to Reviewer 7tHG, two key results that we use in Theorems 4 and 5 are that via duality, 1-Wasserstein distances can be written as a supremum over Lipschitz functions, and the fact that Lipschitz functions on the hypersphere may be approximated by functions in $F_2$ at a certain rate depending on the dimension.
>
> * **“Furthermore, the switch from F_2 to Ftilde_2 is not well motivated - what do we learn about F_2 by looking at Ftilde_2 instead?”**
>
> We will improve the clarity of this part. We will try to make the introduction of the space $tildeF_2 $ clearer. While $F_2$ can be seen as the RKHS with kernel $k(x,y) = \int_{\mathbb{S}^{d}} \sigma(\langle \theta, x \rangle) \sigma(\langle \theta, x \rangle) d\tau(\theta)$ where $\tau$ is the uniform measure over the hypersphere (lines 101-102), $tildeF_2$ is the RKHS we obtain when we replace $\tau$ by the probability measure $tilde \tau$, which has density described in lines 255-256.
> That is, $F_2$ and $tildeF_2$ are both fixed-kernel spaces, and they differ in the weighing measure of the kernel. The fact that they behave similarly is exemplified in Figure 3. $tildeF_2$ is introduced because in Theorem 5 the argument that makes the connection with the sliced Wasserstein distance requires the base measure of the kernel to be $tilde \tau$. It does not work with the uniform measure $\tau$, which does not mean that the result is false.
> Since the experiments are a validation of the theory, we know with certainty the dependency in the degree $k$ of the Legendre polynomial with the dimension: it is given by Theorem 2 for the IPMs and Theorem 3 for the Stein discrepancy (a tight result in the former case; a bound in the latter). As mentioned in lines 170-171 and 214-215, the exponential separation in $d$ takes place when $k$ is of the order of $d$.
>
> * **“You mention setting k=d near the discussion of the implications of the theorem and later in the experiments set k constant for all d - discussion would help.”**
>
> We were constrained in the values of $k$ and $d$ that we could choose for these validation experiments. As we explain in our second answer to Reviewer tnLw, when the $F_2$ IPM is small, which is the case when $k$ and/or $d$ are small, we need a high number of samples from the distributions $\mu_d, \nu_d$ to make the statistical error smaller than $d_{B_{F_2}}(\mu, \nu)$ and get a good estimate. In Figure 1, we state that we use 6M samples from each distribution to obtain the plots, but this is a typo that will be corrected; in fact we use 4400M samples, and we are still barely able to make the theoretical ratio fall within the error bar of the empirical ratio for dimension 16. This takes >10 hours using 10 different nodes in a cluster (one for each of the 10 repetitions). If we were to validate our formula in the setting $k=d$, we would have to restrict ourselves to lower dimensions because otherwise the number of samples required would be unmanageable.
>
> * **“Dimensionality - In sections 3, 4, and 7, the inputs are considered to lie in a subset K of the (d+1)-dimensional space (l. 114), and the paper considers statistical distances between distributions \mu and \nu over K, for two special cases: K=R^d \times {1} or K=S^d. However in sections 5 and 6, the distributions \mu_d and \nu_d are defined over S^{d-1}. Might be useful to keep it consistent”**
>
> The reason that in Sections 3,4 (definition $F_1/F_2$ spaces and $F_1/F_2$ IPMs) and 7 (Relation to Sliced Optimal transport)  we assume $K$ is a subset of $\mathbb{R}^{d+1}$ is to cover the case $K=\mathbb{R}^d \times \{1\}$ which corresponds to neural networks with input in $\mathbb{R}^d$ and bias term. Since $\mathbb{R}^d \times \{1\}$ is a subset of $\mathbb{R}^{d+1}$, to keep a common framework we chose $K = \mathbb{S}^{d}$. Note that these sections are general and not primarily based on spherical harmonics and don’t present separation results.
> However, the results of Sections 5 and 6 are concerned with constructing measures on hyperspheres for which we have separation between $F_1$ and $F_2$,. Thus it was more natural to use hyperspheres $\mathbb{S}^{d-1}$ contained in $\mathbb{R}^{d}$, because this is the notation that is typically used in works that deal with spherical harmonics that are at the heart of the construction (e.g. Atkinson and Han [2012]). We understand that an alternative notation would be to use $\mathbb{S}^{d}$ everywhere.
>
> * **“l. 178: define the acronym EBM, and add the relevant references again”**
>
>  EBMs = energy-based models. We will use the full name.
>
> * **“line 171 it seems like eq (6) should read eq (5)”**
>
>  Yes, this is a typo. Thank you for catching it.
>
> * **“eq (33) is the same as (6) - was a bit confusing in the main text”**
>
> Another typo. Thanks!

---

> > ### Comment · Reviewer_e39f · 2021-08-23
> > **Thanks for addressing my comments, I will reply to the main points.**
> >
> > - “For example, something that immediately comes to mind is that you could remove/project out the high frequency components and see if the FID without feature learning then gets better”
> >
> > Let me clarify and apologies that I essentially did not finish describing what I meant - I was thinking about a new artificially created image dataset (that might not look natural) as the ground truth dataset that has high frequency components projected out. In that case it’s not relevant to have a small FID score compared with the original image dataset. On that new artificially created image dataset you would then try out F2 generative modeling and see if it can achieve smaller FID score than when you try to do F2 generative modeling on the original dataset.
> >
> > For this experiment I’m not sure how Figure 7 of the mentioned paper is relevant?
> >
> > - “Furthermore, the switch from F_2 to Ftilde_2 is not well motivated - what do we learn about F_2 by looking at Ftilde_2 instead?”
> >
> > - “Section 7 and appendix generally seem relatively underdeveloped - the relevance of assumptions and implications of Theorem 5 should be directly discussed and interpreted in order for the reader to know what to focus on while parsing it and perhaps the proof.”
> >
> > I understand that tildeF_2 is an RKHS and that it is necessary for the proof technique to go through. Also thanks for the explanations re Sec 7 - my point was really that more discussions along those lines would be necessary to make this section accessible and insightful for the average reader (primarily not that I was asking for a discussion for myself). Even the first paragraph of Section 7 is rather abstract and unclear (“understanding how these metrics relate to each…”? “work in a similar fashion”) - what’s the purpose of the Section i.e. *why* do I care that "the F1 and F2 IPMs are equivalent to the max-sliced and sliced 1-Wasserstein distances up to a constant power respectively"? Why do I need to learn about bounds of F1, F2 IPM via sliced Wasserstein? And how does it relate/fit to the rest of the paper and story? Now this is not a question I’m asking you to answer to myself but that would be good to spell out more prominently for the reader. Also perhaps a sneak peak of the result (i.e. that they are in fact essentially equivalent) that supports the relevance.
> >
> > Similarly the current comments in the paper regarding Ftilde, e.g. “related class”, “we can switch from uniform to alternative measure” - sure we can switch but why is Ftilde relevant besides the fact that you can show what you want to show … The discussion you provided is addressing the question to some extent, in particular that they seem to behave similarly in some experiments (would I expect that to hold for other k,d and activations that are not plotted in Figure 3? why?). Can this fact (Ftilde and F behaving similarly) be shown (mathematically or experimentally) to hold in more generality than just in the experiments of Figure 3? I’m not sure what you mean by “It does not work with the uniform measure  τ, which does not mean that the result is false.” and how the following relates to my question.  I don’t think I claimed a result was false :)
> >
> > Paragraphs that add motivation and discussions like this (and similarly of the Ftilde part) would require more space - I'm curious how are the authors planning to allocate space in that case? What are the plans in terms of writing for Section 7? E.g. move Stein discrepancy part to the appendix since it isn’t tight anyway?
> > If the authors can be more concrete about changes in the writing of the paper (i.e. which paragraphs they would add where) that make the paper more intuitively accessible via discussions and less focused on purely stating equations, and it makes sense to me, I’m happy to raise the score. My feedback is having the amount of impact of the paper in mind, that could reach many more people if the presentation bridged the link to the more practically minded but potentially theoretically interested ML community better.

---

> > > ### Author Response · Authors · 2021-08-24
> > > **Thanks and response to comment**
> > >
> > > We thank the reviewer for their detailed response and their constructive comments. We address below their main questions and hope that our response on how we plan to incorporate the feedback will be to their satisfaction and  improves their scoring of the paper.
> > >
> > > > “I was thinking about a new artificially created image dataset [...] on the original dataset.“
> > >
> > > We appreciate the idea, and would like to thank the reviewer for the clarification. Indeed, it would be interesting to see whether for this artificial dataset the FID score of the $F_2$/kernel generative modeling is better than for the original dataset. It would also be worthwhile to check if the $F_2$ generative modeling is competitive against $F_1$ generative modeling on the artificial dataset, when the frequency cutoff is low enough. As we mentioned in our answer, our paper is of theoretical nature, and we leave this for future work.
> > >
> > > > “why do I care that "the F1 and F2 IPMs are equivalent to the max-sliced and sliced 1-Wasserstein distances up to a constant power respectively"? Why do I need to learn about bounds of F1, F2 IPM via sliced Wasserstein? And how does it relate/fit to the rest of the paper and story?”
> > >
> > > The results of Section 7 are different in nature from those of Sections 5 and 6 (for starters, note that in Sections 5 and 6, $K$ is the hypersphere and in Section 7, $K$ is {1}$\times${$ x \in \mathbb{R}^d | \|x\|_2 \leq 1$}). In Section 7, the high-level goal is to get a quantitative understanding of how strong feature-learning and fixed-kernel IPMs are in relation to other metrics which have been studied more extensively (namely, sliced Wasserstein distances). We believe that the link may be interesting to researchers in both the areas of generative modeling and optimal transport, and Reviewer 7tHG also appreciates the unifying aspect of the contribution.
> > >
> > > We will expand the motivation of this section from the use of Wasserstein distance with learned cost function (Sinkhorn GAN) that can be seen as using a max sliced wasserstein (spiked wasserstein to be more precise, where the feature map use a learned low dimensional linear map). Similarly, average sliced Wasserstein has been used in the  context of generative modeling; we will add all appropriate citations to these works. Section 7 serves two purpose : First  by showing that the $F_1$ IPM is equivalent as a metric to the max sliced Wasserstein, and that the $F_2$ IPM is  equivalent as a metric to the average sliced Wasserstein, our separation results between $F_1$/$F_2$ IPM translate to separation results between max and average sliced Wasserstein. The second purpose is to deepen the understanding of $F_1$ and $F_2$ by linking them to sliced optimal transport .
> > >
> > >
> > > > why is Ftilde relevant besides the fact that you can show what you want to show
> > >
> > > Indeed, we introduce $tilde \mathcal{F}_2$ because it is the appropriate RKHS to obtain the link with sliced Wasserstein shown in Theorem 5. One might take a different perspective in this matter: the space $F_2$ and the kernel $k$ was introduced by [Bach, 2017], but it is not tailored to the case of neural networks with bias term. Our result can be seen as a theoretical argument in favor of $tilde \mathcal{F}_2$ and $\tilde{k}$, at least when one wants to use a fixed-kernel discriminator. It would be interesting to study whether $tilde \mathcal{F}_2$ presents any advantage over $F_2$ in the regression problem studied by [Bach, 2017].
> > >
> > > > “I'm curious how are the authors planning to allocate space in that case?”
> > >
> > > We will have enough space for additional explanations, because: (i) As per your suggestion, we will remove lines 189-196 and replace them by a single sentence mentioning KSD, because the explanation is not needed for the rest of the paper and we will move this to appendix (ii) if the paper gets accepted, we will be able to use an additional page to incorporate the feedback in the main text, according to the instructions in the Call for Papers: **“If your submission is accepted, you will be allowed an additional content page for the camera-ready version.”**
> > >
> > > > “If the authors can be more concrete about changes in the writing of the paper [...] I’m happy to raise the score.”
> > >
> > > We thank the reviewer again for their constructive comments that will improve the paper clarity.  We will incorporate your feedback and  modify the structure in Section 7 as follows:
> > >
> > > * **Introductory paragraph:**  We will improve the motivation as discussed earlier.  “$F_1$ and $F_2$ IPMs measure differences of densities by slicing the input space and then maximizing (resp. averaging) the appropriate quantities. Max-sliced and sliced Wasserstein distances work, which have been studied by several works, work in an analogous fashion; one projects the distributions onto one-dimensional subspaces, and then maximizes or averages over the subspaces. The goal of this section is to show that $F_1$ IPMs are equivalent to max-sliced Wasserstein distances up to a constant power, while sliced Wasserstein distances are similarly equivalent to a fixed-kernel IPM with a kernel that is slightly different from the $F_2$ kernel. These bounds are helpful to get a quantitative understanding of how strong feature-learning and fixed-kernel IPMs are, and provide a novel bridge between sliced optimal transport and generative modeling discriminators.“
> > >
> > > * **Subsection 7.1:** We leave it as is (correcting the typo “measures in $\mathcal{P}(K \times K)$”). We add a last paragraph in which we remark that the one-dimensional subspaces along which we project may be regarded as features, and that from this perspective max-sliced Wasserstein distances perform feature-learning because it selects the Wasserstein distance along the most different direction, while sliced Wasserstein distances resemble fixed-kernel discriminators in that they average over all the directions. This will provide a smooth transition to our theorems, which formalize this intuition.
> > >
> > > * **Subsection 7.2:**  We remove lines 242-244 because they are not clear. We simplify the statement of Theorem 4 by removing equation (15) and leaving just the inequalities in line 250, which convey the message of the theorem. We change the introduction of the space $tilde \mathcal{F}_2$: we first introduce the kernel $\tilde{k}$, we keep Proposition 2 to link $\tilde{k}$ and $k$ and then we remark that the only difference between the two kernels is the base measure for the features: while for $k$ it is the uniform measure, for $\tilde{k}$ it is the measure $tilde \tau$, for which the mass is more concentrated around the two poles in the direction of the bias term (as we deduce from the factors $(1-t^2)^{-1/2}$ vs. $(1-t^2)^{(d-1)/2}$). We introduce the space $tilde \mathcal{F}_2$ as the RKHS for $\tilde{k}$, and then introduce the $tilde \mathcal{F}_2$ (equation 16). We simplify Theorem 5 by removing equation 17 and leaving just the inequality in line 265. We add the comment that the switch from $\tau$ to $tilde \tau$ is to make the proof go through, the one we make in the answer to “why is Ftilde relevant besides the fact that you can show what you want to show ''.

---

> > > > ### Author Response · Authors · 2021-09-01
> > > > **Follow-up**
> > > >
> > > > Dear Reviewer e39f,
> > > >
> > > > We would love to hear from you whether we have addressed your concerns in the previous reply. And we are happy to clarify further if there are any remaining questions. Thank you!

---

### Decision · Program_Chairs · 2021-09-27

**Decision:**

Accept (Oral)

**Comment:**

The authors present interesting theoretical analysis with wide-reaching implications; they are able to exhibit sequences of distributions which are distinguished by F1 integral probability metrics (based on an infinite width neural network function class that performs "feature learning") but not F2 integral probability metrics (a function class that does not perform "feature learning"), and in doing so they provide theoretical support for the notion that feature learning is an important property for discriminators (e.g. an argument against the use of "MMD GANs", which have previously appeared in NeurIPS).  The analysis is extended to cover Stein discrepancy, again based on F1 and F2.  All reviewers agreed that the paper was well-written and correct.  I believe these results will be of significant interest to the participants of NeurIPS.